# Classifying Children’s Behaviour at the Dentist—What about ‘Burnout’?

**DOI:** 10.3390/dj11030070

**Published:** 2023-03-02

**Authors:** Christopher C. Donnell

**Affiliations:** 1Academic Unit of Oral Health, Dentistry and Society, School of Clinical Dentistry, University of Sheffield, Sheffield S10 2TA, UK; c.c.donnell@sheffield.ac.uk; 2Department of Paediatric Dentistry, Charles Clifford Dental Hospital, Sheffield Teaching Hospitals NHS Foundation Trust, Sheffield S10 2SZ, UK

**Keywords:** paediatric dentistry, patient communication, behaviour management, burnout, dental anxiety, dentist–patient relationship, patient-centred care

## Abstract

In children and young people, complex and prolonged dental treatment can sometimes be met with resistance despite previously successful treatment appointments. While this has traditionally been referred to as a ‘loss of cooperation’ or ‘non-compliance’, these children may actually be experiencing ‘burnout’, of which many may have the potential to recover and complete their course of treatment. Burnout has been defined as “*the extinction of motivation or incentive, especially where one’s devotion to a cause or relationship fails to produce the desired results*”. Traditionally, burnout is experienced by those who deliver services rather than be in receipt of a service; however, the burnout concept proposed in this paper explores it as an alternative perspective to other dentally relevant psychosocial conditions and should be considered when employing appropriate behaviour management techniques and coping strategies for paediatric patients. The intention of this paper is not to establish firm grounds for this new concept in healthcare, but to start a discussion and motivate further theoretical and empirical research. The introduction of the ‘burnout triad model’ and the importance of communication aims to highlight the tripartite influence of patients, parents and professionals engaged in the central ‘care experience’ and underlines the belief that early recognition and management of potential signs of burnout may help reduce the likelihood of those involved developing the condition.

## 1. Introduction and Background

Over the last 25 years, the concept of ‘burnout’ in dentistry has been attributed solely to dental professionals, with Maslach and Jackson defining burnout as a ‘syndrome of emotional exhaustion (EE) and cynicism that occurs frequently among individuals who do “people-work” of some kind’ [1]. Freudenberger, the psychoanalyst who first coined the term ‘burnout’ in 1974, defined it as “*the extinction of motivation or incentive, especially where one’s devotion to a cause or relationship fails to produce the desired results*” [2]. In 2019, the World Health Organisation (WHO) officially recognised burnout as an occupational phenomenon, advising it was brought about by “chronic workplace stress that has not been successfully managed” [3]. In the same year, a survey of dentists in the United Kingdom (UK) found high levels of stress and burnout alongside low well-being, with general dental practitioners (GDPs) particularly affected [4]. Work stress and burnout among dental hygienists is also prevalent, with a recent study noting that one-in-eight dental hygienists felt emotionally exhausted from work and, when compared with other professionals, were relatively negative about the variety of tasks they find in their work [5]. Internationally, in the United States, more than half of US physicians are now experiencing professional burnout [6], and there are similar findings from studies in the Middle East [7], Europe [8], South America [9] and Asia [10].

With regard to the management of burnout in dental professionals, Newton et al. [11] recommended interventions which address a broad range of outcomes, including the experience of stress, coping mechanisms, behavioural change and exposure to stressors. A recent systematic review by Plessas et al. [12] recommended that mental well-being awareness be put at the centre of dental education and the workplace—they suggest primary-level interventions which can be implemented in UK dental practice, such as those recommended by Basson [13]: stress management, cognitive-behaviour training, mindfulness-based-stress reduction (MSBR), mindfulness meditation (MM), rapid relaxation (RR) and narrative counselling, in addition to additional preventive factors such as social and peer support, assertiveness training, physical exercise and progressive muscle relaxation (PMR). Morse et al. [14] advised that the use of multiple strategies may be advisable instead of relying on one technique.

Research suggests that “feeling as the other person feels”, i.e., exhibiting empathy, plays a vital role in clinical relationships [15]. Empathy consists of moral, emotive, cognitive and behavioural components and is associated with compassion, thoughtfulness, attentiveness and caring [15,16]. A professional’s reduced ability to exhibit empathy has been typically associated with compassion fatigue, of which burnout is a major component [16]. Compassion fatigue is a specific form of burnout from a professional’s deep investment in aiding others [17,18]. As burnout is traditionally experienced by those with a responsibility for delivering care for others, the construct of ‘moral injury’ is also often associated with accounts of burnout [19]. Moral injury is defined as “the mental, emotional, and spiritual distress someone feels after perpetrating, failing to prevent, or bearing witness to acts that transgress deeply held moral beliefs and expectations” [19,20]. The demands of the dental educational process instils the desire to provide the best clinical care to patients; however, once professionals enter the ‘real world of dentistry’ some are confronted with business-oriented, profit-driven practices, and have reported facing conflicting interests of providing quality care or the practice’s profitability [21]. It has further emerged in the healthcare discussion recently because of the challenges placed on healthcare workers and systems in the context of the COVID-19 pandemic [22].

While the majority of studies surrounding burnout and dentistry focus predominantly on dentists, dental care professionals and dental students, there is a distinct paucity of any exploring burnout from the patient perspective [23,24]. It was only in 2018 that Bain and Jerome [25] first proposed the concept of ‘patient burnout’, asserting that increasingly complex dental treatment and high and/or unrealistic expectations, alongside a focus on treatment rather than care (with more emphasis on performing procedures than diagnosis and treatment planning), all contribute to the development of an emotionally exhausted, burnt-out patient. Similar musings were proposed by Hoover [26] in the early 1980s, prior to official recognition of ‘diabetes burnout’, where he postulated that the chronic stress and frustration of diabetes management led to patients experiencing burnout being misdiagnosed as ‘non-compliant’. Although ‘burnout’ is a term used widely across many industrialised countries, it is often poorly understood and stigmatised, and therefore is not always managed effectively [27].

Children and young people (CYP) are not exempt from experiencing burnout. Academic burnout, sometimes termed ‘school burnout’, is considered “*an extension of career burnout as students’ routines may include overly structured activities, such as attending class and submitting assignments*” [28,29]. Academic burnout is typically applied in the context of older school students; however, it may also affect young children as well [30]—following the return to face-to-face education and unparalleled periods of virtual learning from home, teachers and schools have started to focus on making up for learning loss, which may result in increased academic intensity that younger students have little experience of, with Wu et al. [31] citing school pressure, peer groups and school engagement as risk factors for academic burnout.

Psychological conditions, such as depression and anxiety, are among the most common barriers associated with dentistry and have considerable impact on it. The literature suggests that although these conditions are interconnected, they are different and robust constructs [32]. Depression, for example, is not the same as burnout. They have different diagnostic criteria with different management strategies; while burnout is reported to improve with a break or time away, depression does not and tends to pervade every domain of a person’s life [27].

Much of the literature around burnout refers to it as a condition specific to the workplace. As burnout is not just limited to adults, the author proposes that many children suffering from dental burnout may be commonly mislabelled as ‘uncooperative’, ‘non-compliant’ or even ‘unmotivated’. As such, this paper builds on the work of Bain and Jerome [25] and proposes burnout as a separate psychosocial entity to consider in paediatric patients. The intention of this paper is not to establish firm grounds for this new concept in healthcare, but to start a discussion and motivate further theoretical and empirical research, by helping dental professionals recognise and identify potential warning signs, alongside tentatively proposing strategies to prevent, minimise and manage dental burnout in children and young people.

## 2. Recognition and Identification

### 2.1. The Dental Burnout Concept

Just like young Alexander in Judith Viorst’s children’s novel of the same name, any child can have a “*Terrible, Horrible, No Good, Very Bad Day”*, especially when anxious children know they are going to the dentist. While a child’s behaviour is typically reflective of their mood, their emotions and their maturity level, one ‘negative’ interaction at the dentist should not be interpreted as a complete loss of cooperation. Before introducing the concept of dental burnout and how it may relate to cooperation, it is first important to establish how cooperation may be gauged clinically in terms of patient behaviour.

The British Society of Paediatric Dentistry (BSPD) and the Royal College of Surgeons of England propose that children’s behaviour may be characterised into one of three categories: ‘cooperative’, ‘lacking cooperative ability (sometimes referred to as ‘pre-cooperative’)’ or ‘potentially cooperative’ [33]. The BSPD advise that awareness of the clinical features of these distinctive categories may be helpful to enable appropriate behaviour management and treatment planning (Table 1) [34].

While general signs of burnout may look the same in children, adolescents and adults, CYP who lack cooperative ability, including those for whom attempted acclimatisation is unsuccessful (due to, e.g., dental fear and anxiety [DFA]), are unlikely candidates to experience dental burnout. Instead, this paper hypothesises that burnout may be experienced by cooperative and potentially cooperative patients who manifest signs and symptoms despite multiple previous *successful* treatment sessions in the *same* course of treatment, i.e., not a loss of cooperation after one visit or even during the first visit. Clinicians need to be mindful that cooperative behaviour may change across sequential treatment appointments. As such, the dental burnout construct may result in part from two types of behavioural learning: operant conditioning and learned helplessness [35].

When children encounter recurring pervasive or challenging experiences, such as those which have been reported during expansive courses of dental treatment [36], they often experience a stressful feeling of helplessness and loss of control that may engender a lack of motivation and effort in continuing the pursuit of previous goals, such as the completion of treatment [37]. This theoretical domain was termed ‘learned helplessness’ in the 1960s by Seligman and Maier [38] and is actually based on operant conditioning learning experiments with dogs. Despite a long history of theoretical and empirical research into motivation science since, however, a comprehensive framework into the causal factors and mechanisms involved in motivation psychology is still lacking [37]. Both in daily life and in research, burnout is often regarded as the depletion of a finite resource of physical and/or psychological energy; the term itself lends to the concept that an individual has run out of a finite source of energy (or resilience) [39,40], and was been popularised by the Baumeister’s ‘Ego Depletion Theory’ [41,42], which proposed that willpower draws upon a limited pool of psychological resources that can be used up. The theory has, however, been criticised recently on a number of theoretical [43] and empirical grounds [39,44]. The burnout concept theorised in this paper is based on work by Boddez et al. [39], who broadly define learned helplessness as ‘*the behavioural effects of a lack of (positive and negative) reinforcement*’, and suggested combining the operant, stimulus-based domain of learned helplessness with goal-directed stimuli and their influences on behaviour and perceived control. They propose that emotional dysregulation may result from mismatches in expectations, where goals that seem unattainable subsequently lose value [45], such as if dental treatment over time is perceived as ‘punishing’, and afterwards induces a belief that further effort will not result in attaining their goals [37]. Although initial behaviour at the new patient appointment can serve as useful foreshadowing into cooperation at future appointments, signs of dental burnout in children can slide under the radar and may not become noticeable until sometime later when there are clear changes to a patient’s behaviour [46].

### 2.2. Signs and Symptoms of Dental Burnout

As alluded to above, the main stay of this concept is that these signs and symptoms may manifest following multiple previously successful and similar appointments. The literature suggests that awareness of the following signs and symptoms hypothesised to be associated with dental burnout may help prevent patients from feeling mentally defeated, overwhelmed and closing off [25,47]:Loss of care and motivation—the child may no longer present with the same enthusiasm as previous treatment appointments; they may become less communicative, including blunt or one-word replies, and no longer maintain eye-contact;Avoidance behaviours—the child who previously attended treatment without concern may attempt to avoid it completely by, e.g., refusing to come in from the car or waiting room, or may refuse to come at all and want to stay at home/school;Worry, anxiety and fear—the previously calm and composed child may be experiencing an increase in fear, worry or anxiety, particularly if they are approaching more invasive aspects of treatment such as exodontia—when these feelings outweigh a child’s ability to cope it may result in tears, feeling sick (including stomach aches and headaches), shaking, sweating and freezing;Attitude shift and emotional changes—the child’s attitude may shift from being positive, open and honest to a predominantly negative mindset—they may be less tolerant to minor setbacks such as being kept waiting a few minutes past their anticipated appointment time or recurrent loss of orthodontic separators prior to preformed metal crown (PMC) placement;Concentration—the child may become easily frustrated with situations or aspects they did not previously get triggered by, such as lost fillings or administration of local anaesthetic; in addition, they may find it hard to keep concentration and exhibit annoyance when required to sit for longer appointments such as during root canal treatment.

### 2.3. Likely Candidates for Dental Burnout

Childhood experiences can have profound and lifelong effects. Although there is an abundance of research investigating the long-term effects of childhood dental fear and anxiety, little consideration has been given to exploring burnout from the patient perspective [23,24,25,48]. Burnout in medical professionals is pervasive and well-studied; however, even in medicine, little is known about the prevalence and severity of burnout among patients. The medical literature suggests some similarities to dentistry, however, as patients with complex and lengthy treatment plans, especially those requiring multidisciplinary input from two or more specialties, are particularly vulnerable to burnout [49]. It is the author’s experience that the following groups may be at particular risk of dental burnout:Limited dental experience—if a child’s only dental experiences have involved check-ups or basic operative care (typically without local anaesthetic), their lack of ‘*dental sophistication*’ can result in limited treatment stamina and an inability to cope with longer and, on occasion, uncomfortable appointments further along a complex treatment plan [25];Dental trauma—in addition to the initial physical trauma itself, e.g., tooth avulsion, the experience of immediate and short-term management is often painful and frightening for a child and can significantly impact them on an emotional and psychological level; significant long-term monitoring is necessary, particularly in patients with a developing dentition, alongside knowledge of risks such as loss of vitality and the need for more invasive treatment such as root canal therapy [50];Medical trauma—medical trauma may occur in children as a response to single or multiple medical events; it refers to the psychological (and sometimes physiological) response of a child to pain, injury, serious illness, medical procedures and invasive or frightening treatment experiences in the medical setting, such as those involved with childhood cancer—young children are still developing their cognitive skills and process information differently, hence they may associate dental pain with punishment and believe they did something wrong. This could lead to burnout during invasive dental treatment, despite previous successful appointments [51];Developmental dental defects (DDD)—developmental defects of the enamel and dentine are lifelong conditions that require multidisciplinary input as well as short-term and long-term management that can be increasingly challenging in young children. Pain, sensitivity and aesthetic concerns are commonplace, alongside a reduced strength and integrity of the bond of enamel to composite that often results in failed restorations. Molar Incisor Hypomineralisation (MIH), for example, with a global prevalence of 14.2%, has a significant impact on both patients and dentists and has been shown to negatively impact a patient’s oral-health-related quality of life (OHRQoL)—a recent systematic review by Jälevik et al. [52] found that already-restored MIH molars remain within short re-treatment cycles; sensitivity becomes problematic when it hinders the possibility of obtaining sufficient pain control; and, consequently, behavioural management problems arise due to dental fear and anxiety related to the pain experienced by patients during multiple treatment appointments. In addition, the burden of other chronic developmental conditions such as amelogenesis imperfecta (AI) or dentinogenesis imperfecta (DI) may also be associated with significant medical co-morbidities, further increasing the risk of burnout due to possible previous medical trauma [53];Dental fear and anxiety (DFA)—DFA is common in CYP, with an estimated prevalence between 6% and 20% in those aged 4–18 years old [48]. Potentially cooperative children, especially those who exhibit ‘timid’ and/or ‘tense-cooperative’ behaviour, are at increased risk of burnout, should they be exposed to invasive and uncomfortable procedures without adequate preparation and coping strategies;Re-treatment—children who require extensive re-treatment because of failed or failing treatment are increasingly susceptible to burnout. This includes changes as a result of failed complex treatment or further trauma, as well as recurrently debonding restorations such as those associated with AI. Patients requiring re-treatment may be close to burnout already, due to ongoing frustrations with previous protracted treatment in addition to unanticipated problems which require subsequent further treatment;Personal circumstances—children who undergo a significant change in circumstances during a long treatment plan are also at risk of burnout. These are usually external to the dental setting and may be due to finding transitions into school extremely stressful, continuous exposure to stressful or traumatic events, struggling with external changes or being extremely driven to excel at school exams [47,54].

## 3. Minimising and Managing Dental Burnout

It would be impossible to anticipate every individually small but cumulatively significant manifestation of dental burnout in children. Instead, the WHO support the proactive promotion of positive experiences in childhood, as they set the foundation for optimal childhood development and subsequent flourishing in adulthood [55]. This stance aligns with research by Bethell et al. [55], who advise that attention should be given to the creation of positive experiences which both reflect and generate resilience within children, families and communities, as they have the ability to mitigate the fallout from negative childhood events. As such, the author suggests the following strategies which may help to minimise the hypothetical risk of dental burnout, as well as other overlapping psychosocial constructs, during dental treatment in children.

### 3.1. History Taking

While it may not be their first visit to the dentist, a child’s first visit to *you* is key when assessing their behaviour in *your* dental setting. Although some children are robust and tolerant in stressful situations, others may appear vulnerable and require more time and attention to feel at ease and subsequently cooperate with dental treatment [34]. Cooperation levels at the initial visit may also be greatly dependent on why the child believes they are there. Take time to give reassurance and explore any anxiety they may be exhibiting—some of the behaviour may be related to the child’s perception of the dental surgery, or even from a prior medical experience. These external factors may be revealed if appropriate questions are asked concerning all aspects of dental, social and medical history. Use of a ‘*functional inquiry*’, a series of questions posed as a relaxed personal interview, may be advantageous as it helps: elicit dental problems, explore the child’s behaviour (both inside and outside the dental setting), understand the parent’s attitude and assess the potential for patient cooperation [34]. Anxious patients may benefit from established resources such as ‘Message to Dentist’ to help facilitate communication between anxious children and the dental team [56]. A thorough history may provide good insight into whether the child is at risk of burnout further down the treatment line, in addition to those children for whom attempting chairside treatment would be futile. A useful example is provided by Locatelli [51]:


*“…his behavior [sic] was at times totally dysregulated. While taking a health and medical history, they learned that the child, David, had endured many painful, frightening, and invasive medical procedures beginning at 21 months of age…”*


Shared understanding and open communication will improve trust. Some healthcare professionals typically view a lack of cooperation as a moral failing; however, we may have simply failed to see the wider picture and the many steps that led the child to be in that position. It is essential to recognise that some children present with a history of lost trust opportunities or disparities that they have shouldered through previous healthcare encounters, and it is up to us to discover those while taking a patient history.

### 3.2. Communication

All behaviour is communication. When a child begins to appear less cooperative, they are trying to communicate that something is up. Sometimes they are frustrated, or tired or even overstimulated. Maybe they are mad, or lonely or simply scared. Children are not always able to communicate exactly what is going on; however, it is our job to figure out what the issue is. De-escalate the situation and approach it with calmness. Our reaction to their behaviour and our interactions with them direct how they will react and behave in the future. Communication always needs to be honest to ensure there is a reciprocal feeling of trust throughout the course of treatment and beyond. It is very hard to regain a child’s trust once you have lost it.

It is important to avoid inaccurate communication: do not promise treatment completion within unrealistic timeframes; outline the possibility of delays; ensure risks as well as benefits are explained to patient and parent alongside all reasonable options; and keep talking throughout treatment—tell them what you are going to do, discuss what you are going to do (should they have any questions) and on completion, review key points, reinforce positive behaviour and tell them what they will accomplish next time [27]. Such approaches align with the goal-directed stimuli construct associated with the hypothesised burnout concept, on a basis of learned helplessness.

Listening is a cornerstone of communication. Children may communicate their emotional exhaustion using a myriad of the expressions summarised in the previous section and it is important that we recognise the warning signs. If burnout appears to be manifesting and the clinical picture appears relatively stable, it may be wiser to give patients a micro-break from treatment, rather than try to persuade them towards rapid completion [27].

### 3.3. Emotional Exchanges

Each trip to the dentist signifies a type of invisible emotional exchange for a child. Just as the battery in a mobile phone needs to be recharged after continuous use, so does the ‘resilience battery’ in children (Figure 1). The burnout construct hypothesises that a portion of the battery’s charge (resilience) is used each visit, with treatment visits using considerably more than check-up visits, whereas charge may also be topped up following a positive experience. This also lends itself to the theory of latent inhibition, which postulates that successful experiences may counter the effects of a ‘failure’ [46]. When the battery depletes, however, such as following a string of treatment visits, we need to replenish the spent resilience, or risk a ‘low battery’ warning, i.e., burnout. Should we ignore the warning, we arrive at ‘battery flat’, i.e., loss of cooperation, which can be harder (if not sometimes impossible) to come back from with a simple recharge.

Alongside the contribution of positive experiences to replenishing the battery, it may be that correctly timed micro-breaks during extended phases of treatment are required to avoid total loss of cooperation by overwhelming the patient. From a general standpoint, repeated exposure to positive experiences is the most beneficial technique to increase cooperation; however, busy parents may be reluctant to allow the clinician to advance slowly [34].

### 3.4. Continuity of Care

Treatment by the same clinician enables the formation of a strong, dynamic patient–dentist relationship. Familiarity in trusting relationships with healthcare professionals helps to moderate negative experiences and offers children a consistent arrangement that provides support and freedom—it is under these circumstances that children regain a sense of safety, and their misinterpretations of danger can be overcome, navigating away from burnout [51]. Lack of a trusting relationship, and therefore lack of knowledge of a patient’s character, behaviour and clinical limits, presents a real risk of taking an inexperienced patient too quickly from emergency or basic care to ‘advanced’ care such as is seen when a child suffers dental trauma. Just as you would not expect a child who has only ridden a tricycle to cope with a mountain bike, it may be best to take inexperienced patients gradually towards more complex care to minimise potential burnout [27].

### 3.5. Cooperation vs. Compliance

Remember that compliance is not cooperation. If what we strive for as clinicians is a happy and willing patient, then we are talking about cooperation. Compliance relies on a hierarchy of sorts, where the lesser individual must yield. It requires tools which damage the patient–dentist relationship through creating restrictions and requirements that forcefully shape a patient’s choices, inevitably ending abruptly and turning negative when a moment of disagreement arises [57]. Cooperation, on the other hand, provides a semblance of equality and relies on constantly improving the patient–dentist relationship by using tools which strengthen rapport—support, encouragement, listening, trust, respect and negotiating differences. A focus on these approaches will attempt to keep the risk of potential burnout to a minimum.

### 3.6. Behaviour Management Spectrum

As highlighted previously, functional inquiries allow a clinician to gain as much knowledge as possible about their patient, thus providing a relatively reliable estimate of their cooperative ability. As such, they also provide insight into which behaviour management techniques will be most appropriate. Functional inquiries are not limited to new patient appointments; when children have been patients for a long time, situations change, and a periodic history review is in order—the astute clinician keeps patient information up-to-date, and functional inquiries allow a clinician to identify early signs of burnout [34].

While cooperative and potentially cooperative patients have the potential to recover from burnout, there will be some for whom progression along the spectrum of behaviour management may be required. If a microbreak from treatment, for example, does not fully recharge that ‘resilience battery’, the utilisation of conscious sedation techniques such as relative analgesia (RA) may help overcome specific anxieties that are fuelling the burnout. In some instances, general anaesthetic (GA) may be the only appropriate option; however, if restorative treatment (e.g., PMCs, fissure sealants, fillings) can be performed awake, patients requiring exodontia-only GA wait considerably less time that those waiting for comprehensive care.

Pharmacological techniques typically only *manage* rather than *reduce* children’s anxiety; hence, greater effort should be directed toward behaviour and psychological interventions, which have been shown to have better long-term impacts [58]. Behaviour strategies are employed by clinicians on a regular basis and micro-breaks present the perfect opportunity to introduce cognitive behavioural therapy (CBT) into the equation. CBT is a “goal-orientated talking therapy” which aims to help patients manage their problems by generating resilience and changes how they think and behave in relation to their problem; CBT has a strong evidence base for reducing a variety of paediatric anxiety disorders and associated co-morbidities [56].

## 4. The Burnout Triad

Children cannot bring themselves to healthcare appointments. Instead, they are ‘brought’, either willingly or unwillingly, by an accompanying parent or guardian. Uniquely for CYP, this introduces a third party into a traditionally bi-directional professional–patient dyad [27]. While not the chief aim of this paper, the author wishes to briefly hypothesise the concept of a ‘burnout triad model’ (Figure 2), an extension of a recent concept by Martinez-Hollingsworth et al. [49], which reframes burnout and moves it beyond traditional thinking by recognising the pervasiveness of burnout on the focal ‘care experience’ and how it may be influenced by any combination of patient, parent or professional.

The burnout triad model incorporates previously held theories in the literature surrounding professional factors in which provider burnout suggests that, as clinicians, we are exposed to growing numbers of patients undergoing prolonged and/or complex treatment, and thus potentially manifesting signs of burnout. This cumulatively overwhelms the provider, leading to the emotional exhaustion typical of professional burnout, thus developing a vicious circle [27,49]. With regard to paediatric dentistry, however, a 2020 study of occupational burnout among paediatric dentists found that despite the belief that paediatric dentists may be at greater risk of occupational burnout due to the chronic stress associated with provision of paediatric dental care, only nine percent fulfilled their definition of occupational burnout (high emotional exhaustion and depersonalisation) [59].

Central to the care experience of the burnout triad model is communication. A significant prerequisite for child participation in healthcare encounters is effective communication, especially as child participation is critical when providing child-centred care [60,61]. Yuan et al. [62] noted that communication with children in dental settings is potentially more complex as there is a “*prerogative to provide appropriate information to the child for prevention and treatment as well as preparing them for dental procedures*”. Their theoretical hypothesis is that adult behaviours (professional and/or parental) might have an important part to play in child participation in the dental setting, and seeks to unlock the dynamics of the complex communication process that exists in the dental setting [63]. Analysis of the process is challenging given the complexity of managing a communication process involving three active, yet historically unequal, participants (i.e., the patient (child), parent and the professional). This may, however, provide some clarity into how the relationship with the dentist, satisfaction with ongoing communication and changes in the understanding of treatment have an impact. They have carried out seminal work to code, catalogue and define communication behaviours between the triad of dental professionals, children and parents in a clinical setting, which attempts to help clinicians further understand the complexity of the interactions and provide helpful structures to do so [63]. Such work is crucial as, in order for others to be able to explore the triadic interaction in the future, a valid and reliable means of measuring communication behaviours between clinician, child and parent is required [62]. They grouped communication in terms of adult (professional and parental) and child participation, with coded behaviours such as ‘social talk’, ‘praise’ and ‘dental engaging talk’ for professionals; ‘parental facilitation’, ‘joke’ and ‘praise’ for parents; and ‘speech yes’, ‘speech no’, ‘speech other’ and ‘dental answer’ for children. They noted an important association between the time of the professional and/or parental behaviour and child participation, in that the more extended the duration of the consultation, the less likely the communicative behaviours of the adults are exhibited to show participation of the child [64]. This is important as it may imply that initial communication with the child in the dental visit is crucial, with less emphasis on subsequent behaviours later in the consultation to influence child participation. It is imperative to note, however, that cooperation and participation of a child patient are not identical behaviours, although they have been shown to be linked [65].

In addition, the burnout triad model acknowledges parental burnout and the vicarious nature of dental treatment for children and their parents (parental factors); although parents may consent to complex treatment extending many months and multiple visits, factors such as parental anxiety, for example, can have a direct and negative effect on outcomes, as the absence of positive experiences and parental support can be very stressful for children [46,66,67]. In addition, parental presence in the dental surgery has been a heated debate for decades, with diverging opinions. Future research into this evolving area is required.

## 5. Limitations

At present, the construct of dental burnout in paediatric patients remains a hypothesised concept. The classification has not yet been empirically tested or subject to expert debate and consensus. As the reader has already been alerted to, the burnout concept presented in this paper takes some of its founding from other psychosocial conditions; hence, there is likely to be substantial overlap with other known theories and their management. Recent studies raise the question of whether the concept of “learned helplessness” is still timely as it is based on operant learning experiments with dogs; however, there have been many subsequent studies with human participants based on the original concept; hence, the author contends that the transfer of the basic principle across species’ actually strengthens the concept rather than weakens it. The term ‘dental burnout’ may therefore be more relevant to the dental setting and easier for the public to comprehend, as it also encompasses aspects of modern theories such as compassion fatigue, empathy and moral injury. The dental burnout concept provides an attempt to understand children over the longer term in their dental care experience, as the approach is sensitive to the fact that children develop over time and their response to dental care changes—the dental literature is poorly served with such a focus.

## 6. Conclusions

It is not only dental professionals who are at risk of burnout. With this paper, the aim was to consider the concept of dental burnout and its application to healthcare and the context of the paediatric dental patient. The author argues that this may be a useful concept in helping to understand children over the longer term in their dental care experience as they progress as part of a complex tripartite relationship. However, the intention is not to establish firm grounds for this new concept in healthcare, but to start a discussion and to motivate further theoretical and empirical research. This paper hypothesises burnout as a separate psychosocial entity experienced by children and young people undergoing complex and/or prolonged treatment. The acknowledgement of the possible existence of dental burnout as well as consideration of the ‘burnout triad model’ highlights the influence of those involved in the ‘care experience’ and underlines the belief that early recognition and management of potential burnout could reduce the likelihood of patients, parents and/or professionals developing the condition.

## Figures and Tables

**Figure 1 dentistry-11-00070-f001:**
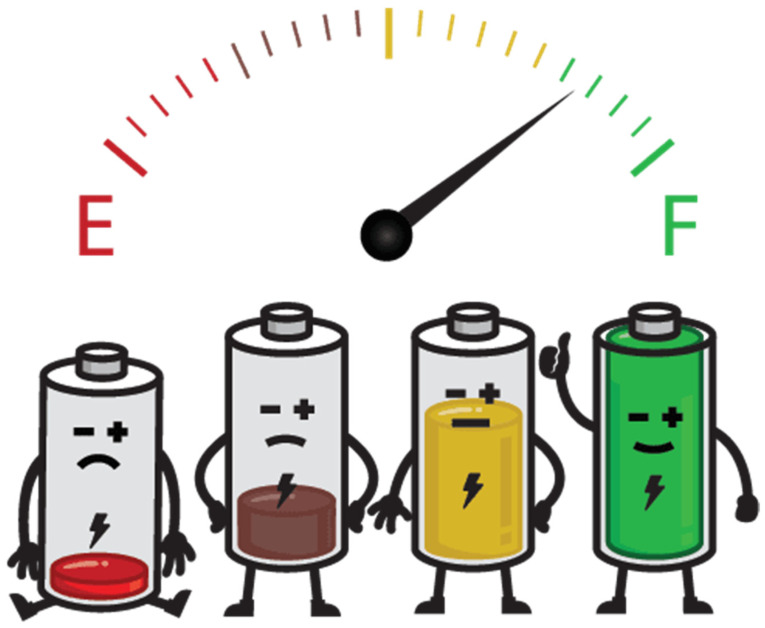
The resilience batteries.

**Figure 2 dentistry-11-00070-f002:**
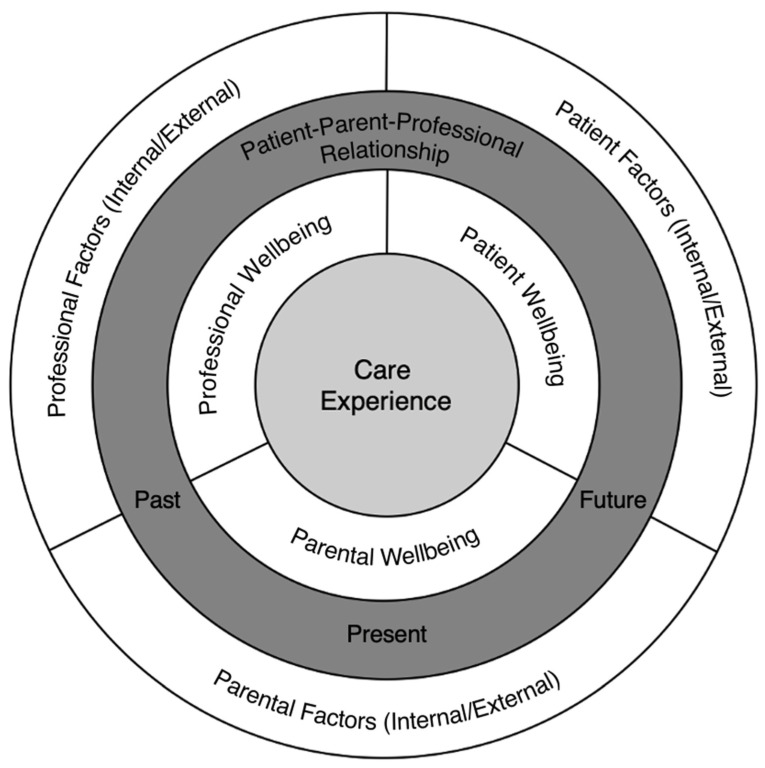
The burnout triad model.

**Table 1 dentistry-11-00070-t001:** Behaviour classifications in children [34].

Category	Description
Cooperative	Reasonably relaxed. Minimal apprehension and may even be enthusiastic. Acceptance of treatment, at times cautious, willingness to comply with the dentist, at times with reservation but follows the dentist’s directions. Can be treated by a straightforward, behaviour-shaping approach. When guidelines for behaviour are established, these children perform within the framework provided.
Lacking cooperative ability (Pre-cooperative)	Very young children with whom communication cannot be established. Comprehension cannot be expected. Lack cooperative abilities usually because of their age. For *pre-cooperative* children, time usually solves the behaviour problems. As they grow older, they develop into cooperative dental patients and treatment is provided with behaviour shaping. Another group of children who lack cooperative ability is those with specific debilitating or disabling conditions. The severity of the child’s condition prohibits cooperation in the usual manner. Although special behaviour guidance techniques are used to allow treatment to be carried out, immediate major positive behavioural changes cannot be expected.
Potentially cooperative	Previously referred to as ‘uncooperative’ or ‘non-cooperative’. This type of behaviour differs from that of children lacking cooperative ability because these children have the capability to perform cooperatively. They have the capacity to cooperate but choose not to—this is an important distinction. They are potentially the most challenging (and most common) patients you are going to meet. When a child is characterised as potentially cooperative, clinical judgment is that the child’s behaviour can be modified; that is, the child can become cooperative. The adverse reactions have been given specific labels for descriptions of potentially cooperative patients, so that potentially cooperative group are further categorised as follows: Uncontrolled behaviour Challenging or defiant behaviour Timid behaviour Tense cooperative behaviour Crying and whining Passive resistance

## Data Availability

Not applicable.

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
