# Peer review of "Classifying Children’s Behaviour at the Dentist—What about ‘Burnout’?"

_dentistry, 2023, doi:10.3390/dj11030070_

Round 1

Reviewer 1 Report (Previous Reviewer 1)

The revised manuscript is much improved and puts forward an interesting idea along with relevant advice for managing paediatric patients. I have only a few minor comments to make to this revision.

Page 2

Depression and anxiety would be better conceptualised as psychological rather than psychosocial conditions.

Page 4

I would remove the term emotional from “two types of emotional learning” since operant conditioning and learned helplessness are behavioural, not emotional theories.

Page 11

Are the items of text referred to as sub-themes not actually codes?

Page 12

You can’t really criticise the theory of learned helplessness and it’s current applicability on the basis of the original experiments with dogs as there have been many subsequent studies with humans based on the concept. Indeed you could say the transfer of the basic principle across species’ strengthens the concept.

Author Response

Reviewer 2 Report (New Reviewer)

The article introduces a novel topic that can open up a very interesting debate for the identification of certain attitudes towards the treatment of child patients. Its identification could open up new paths of treatment or ways of dealing with the treatment of uncooperative children in the dental office.

From the scientific point of view the structure of the article is excellent, with an Introduction that makes the problem addressed perfectly clear.

The rest of the topics dealt with are perfectly explained and particularly the limitations of the study leave the reader with the intention of applying these new concepts, which is quite positive.

I would only add in point 2.3 when talking about DDD, I would also include children with MIH with an explanation of the problem and the difficulty in the treatment of this problem.

I would also congratulate the author of this excellent article.

Author Response

This manuscript is a resubmission of an earlier submission. The following is a list of the peer review reports and author responses from that submission.

Round 1

Reviewer 1 Report

In this interesting article the author proposes a new classification of “treatment burnout” in children who need to undergo complex dental procedures. The ideas are interesting but do not entirely align with current psychological understandings. In parts the author moves to authoritatively making treatment recommendations on the basis of his proposed understanding that are more strongly worded than is warranted given that the classification has not been empirically tested or subject to expert debate and consensus. Although the ideas are interesting it would be important to acknowledge these currently as tentative hypotheses and to consider alternative explanations for scenarios described.

Nowhere in the article does the author address the issue of lack of empathy as a key symptom of burnout or the idea traditionally held that burnout is experienced by people with a responsibility for delivering care to others or the concept of moral injury that is often associated with accounts of burnout.

I would recommend more tentative wording throughout when discussing the concept of treatment burnout in acknowledgement that this is a hypothesis and not an accepted construct.

Page 2

The author presents a theory (concept from Wright about levels of co-operation in children) as if it is a fact and further claims without evidence that “most clinicians, knowingly or not tend to classify children” in this way. I can see no evidence to back up this claim, nor is the theory explained or any evidence presented to back up this statement or to label it as an opinion of the author.

There is also no signposting about why a section headed “signs and symptoms of treatment burnout” is started with behaviour management classifications. The remainder of this section drifts into management recommendations which are made as if the concept of treatment burnout is an established fact.

There should be consideration of alternative theories such as whether such behaviours would fit better with learned helplessness or simple mismatches in expectations about treatment.

Page 3

All of the “signs of potential burnout” could have other explanations. The author needs to make a clearer argument for how and when these signs might be more likely to relate to the proposed construct of burnout. How do the relationship with the dentist, satisfaction with ongoing communication and changes in understanding of the treatment plan have an impact?

Page 4

It is not made clear whether “dental trauma” and “medical trauma” refer to psychological or to physical trauma, or a combination of both. Trauma is a word that is used differently in dental and psychological literature so the precise meaning should be spelled out.

Are the candidates for likely dental burnout not also candidates for the development of learned helplessness and of operant conditioning with treatments over time being experienced as punishing if communication and the relationship are not fully developed?

Following this page guidance is provided which is largely sensible, however it is not clear how this specifically relates to the construct of burnout rather than other known psychological constructs as outlined above.

Page 8

The burnout triad likewise appears to intuitively make sense but seems here to have been rather dropped into the article. An article tentatively considering the construct of burnout and whether it is useful in this setting and going on to also introduce as a hypothesis the burnout triad might be more appropriate.

Reviewer 2 Report

Dear Author

The topic of Ms is very interesting and could help clinician to better understand the behavior of the child and adult patients. Overall, the Ms is well written and structured. However more details could be add in the introduction section and along the text.

Abstract

·       Add the definition or a brief description of burn-out

·       Lines 14-17: this sentence should be rewritten and make clearer.

Introduction

·       Table 1: Table 1 should be built following the author`s guidelines of the Journal

·       Adding the history of burn out and its treatment could be interesting. Author should cite https://doi.org/10.1038/sj.bdj.2016.528, https://doi.org/10.1016/j.mayocp.2015.08.023)

Moreover, Author could also include dental hygienists among the professionals. (https://doi.org/10.1111/j.1601-5037.2005.00130.x)

A summarized chart about the signs and potential treatments of  patient`s burnout could considerably improve the scientific soundness of the Ms. Moreover, this could help the readers to better understand the work.

Section “Burnout Triad Model”: Authors should better explain the Burnout Triad Model explaining each category of Figure 2 and their meaning. Moreover, Author should declare if Figure 2 is original or it was taken from other articles